# Naphthalene-Based Polymers as Catalytic Supports for Suzuki Cross-Coupling

**DOI:** 10.3390/molecules28134938

**Published:** 2023-06-23

**Authors:** Elena S. Bakhvalova, Alexey V. Bykov, Mariia E. Markova, Yury V. Lugovoy, Alexander I. Sidorov, Vladimir P. Molchanov, Mikhail G. Sulman, Lioubov Kiwi-Minsker, Linda Z. Nikoshvili

**Affiliations:** 1Regional Technological Centre, Tver State University, Zhelyabova Str., 33, 170100 Tver, Russia; bakhvalova.es@mail.ru; 2Department of Biotechnology, Chemistry and Standardization, Tver State Technical University, A.Nikitina Str., 22, 170026 Tver, Russia; bykovav@yandex.ru (A.V.B.); mashulikmarkova@gmail.com (M.E.M.); pn-just@yandex.ru (Y.V.L.); sidorov@science.tver.ru (A.I.S.); vp_molt@mail.ru (V.P.M.); sulmanmikhail@yandex.ru (M.G.S.); 3Ecole Polytechnique Fédérale de Lausanne, ISIC-FSB-EPFL, CH-1015 Lausanne, Switzerland

**Keywords:** naphthalene, Friedel–Crafts crosslinking, sulfonation, nitration, palladium, Suzuki cross-coupling

## Abstract

In this work, for the first time, naphthalene (NA)-based polymers were synthesized by one-stage Friedel–Crafts crosslinking. The influence of NA functionalization by -OH, -SO_3_H, and -NO_2_ groups on the polymers’ porosity and distribution of the catalytically active phase (Pd) was studied. Synthesized catalytic systems containing 1 wt.% of Pd either in the form of Pd(II) species or Pd(0) nanoparticles supported on NA-based polymers were tested in a model reaction of Suzuki cross-coupling between 4-bromoanisole and phenylboronic acid under mild reaction conditions (60 °C, ethanol-water mixture as a solvent). These novel catalysts demonstrated high efficiency with more than 95% of 4-bromoanisole conversion and high selectivity (>97%) for the target 4-methoxybiphenyl.

## 1. Introduction

The development of new palladium-containing catalytic systems is of high importance since Pd-catalyzed reactions are considered the most abundant in fine organic synthesis, including cross-coupling reactions [1,2,3,4,5]. Aiming at the sustainability of catalytic processes, a lot of research efforts are devoted to the replacement of homogeneous catalysts by their heterogeneous analogs, which allow easy separation, recycling, and continuous operation, resulting in process intensifications [6,7,8].

It is known that Pd-catalyzed cross-coupling reactions are complex since various processes such as aggregation, dissociation, and leaching of the material can serve as sources of catalytically active species, and some data indicate that the simultaneous active phase can affect the catalyst, leading to a decrease in its activity and selectivity [9,10]. Moreover, in cross-coupling reactions, both Pd^0^ and Pd^2+^ in the starting catalytic presence of several forms of palladium is an advantage [11]. Thus, novel catalytic systems of the Suzuki reaction have been constantly under development [12].

The performance of catalysts can be increased by selecting suitable supports [13]. Hyper-crosslinked aromatic polymers (HAPs), possessing high stability, developed porosity, and diversity of surface functional groups, are considered promising supports and have already been successfully used in heterogeneous catalysis for the immobilization of metal nanoparticles (NPs) [14].

HAPs are characterized by simplicity of synthesis, low cost, and the possibility of using readily available, cheap monomers as starting materials. Friedel–Crafts alkylation of aromatic monomers does not require the presence of specific substituents in the composition of monomers and avoids the use of expensive polymerization catalysts based on noble metals.

The first HAPs were based on polystyrene and were synthesized by co-polymerization of styrene and 5–8% divinylbenzene (DVB). Such HAPs were gel-type polymers and had low porosity. The next generations of HAPs were classified according to the degree of crosslinking and crystallinity. Thus, the second generation had a higher content of DVB (6–12%) and certain porosity. The third generation was prepared in the presence of Lewis acids using a linear crosslinking agent in a solvent, for example, 1,2-dichloroethane (1,2-DCE). In this way, pores with a rigid structure were created, resulting in a surface area of 600–2000 m^2^/g [15].

Currently, the polymers based on polystyrene-DVB (PS-DVB) are the ones most commonly used [16]. Commercial HAPs with various functional groups and high mechanical, thermal, and chemical stability are available [17]. Such amorphous polymers were successfully used as catalytic supports for both Pd complexes and grown Pd^0^ NPs due to their high specific surface area (SSA), developed porosity, and double (hydrophilic-hydrophobic) surface character [18,19,20,21]. The presence of functional groups on the HAPs surface was found to enhance the metal dispersion, while the developed porous structure provided quick diffusion of reagents into the pores and facilitated interaction with active centers.

Tan et al. [13] proposed a simple method for obtaining highly porous polymers from a wide range of readily available aromatic monomers (e.g., benzene or biphenyl) using formaldehyde dimethylacetal (methylal) as a crosslinking agent for condensation of aromatic compounds by rigid methylene bridges using the Friedel–Crafts reaction catalyzed by anhydrous ferric chloride. It was shown that the porous structure and the SSA of the resulting polymers can be regulated by changing the molar ratio of the crosslinking agent to monomers.

Overall, the use of catalytic materials based on HAPs in the processes of fine organic synthesis is promising [18]. However, weak interactions between metal NPs and non-functionalized aromatic supports often lead to leaching and aggregation/sintering of NPs during catalytic reactions. The introduction of functional groups or heteroatoms into the support significantly improves the metal–support interaction and, as a consequence, the dispersion of metal NPs [22]. A number of porous aromatic polymers consisting of nitrogen-rich building blocks (i.e., triazine, carbazole, imine, and amide blocks) were synthesized and used as catalytic supports [23].

One of the ways to obtain functionalized polymers is by using monomers with functional groups. Liu et al. [24] synthesized a series of polymers based on benzyl alcohol via Friedel–Crafts alkylation reaction using methylal as a crosslinking agent and anhydrous ferric chloride as a catalyst. Using phenol as a monomer, hyper-crosslinked polymeric materials were synthesized for CO_2_ adsorption [13].

Another approach to the functionalization of HAPs is the introduction of substituents in phenyl rings after the polymer synthesis. The sulfonation reaction serves as an effective method for obtaining new solid acid catalysts. Li et al. [25] synthesized polymers functionalized by sulfonate groups, which allowed the creation of new heterogeneous acid catalysts. Dong et al. [26] synthesized an efficient solid acid catalyst based on a sulfonated polymer for the conversion of monosaccharides to 5-hydroxymethylfurfural. Thus, mono-functional acid or base catalysts can be synthesized via introduction of acid (-SO_3_H) or basic (-NH_2_) groups into the HAPs’ structure, allowing high stability of the resulting functionalized microporous materials [18].

Naphthalene (NA)-based polymers have been known for more than 50 years. As compared with other monomers (benzene, toluene, and xylene), NA gives a higher polymer yield in the presence of aluminum chloride [27]. Teng et al. [28] developed microporous networks based on NA, 1-naphthol (NL), and 1-methylnaphthalene using the external crosslinker knitting method, the solvent knitting method, and the Scholl coupling reaction. The sample synthesized via Friedel–Crafts reaction while using methylal as a crosslinking agent revealed the highest SSA of 2870 m^2^/g [28].

Thus, NA-based polymers are promising since they possess high SSA, a narrow pore size distribution, and stability. The characteristics of such polymers can be influenced by choosing a monomer, varying the amount of crosslinking agent, and introducing functional groups. However, to our knowledge, the application of NA-based HAPs in catalysis is poorly studied. There is no data on the use of NA-based polymers in the Suzuki cross-coupling process. Moreover, there is no data on the formation of Pd NPs in the environment of NA-based polymers.

Herein, we report for the first time the application of NA-based polymers for the Pd-catalyzed model reaction of Suzuki cross-coupling between 4-bromoanisole (BrAN) and phenylboronic acid (PBA). Several types of polymers were synthesized via Friedel–Crafts alkylation (Figure 1) with NA or NL as monomers at varying levels of methylal content. In addition, one of the NA-based polymers underwent either sulfonation or nitration to find out how the polymer functionalization influences the porosity and hydrophilicity, which is expected to affect the catalytic properties.

Thus, the aim of this work is to show if such properties of NA-based polymers as porosity and hydrophilicity will affect the processes of Pd NPs formation, both during the reduction and in situ during the Suzuki reaction, and also the behavior of the developed catalysts in the Suzuki cross-coupling. Based on our experience in the synthesis of polymeric catalysts of the Suzuki reaction, we suppose that higher hydrophilicity will be able to increase the stability of NA-based catalysts at the recycles.

## 2. Results

### 2.1. Characterization of the Polymers

#### 2.1.1. Liquid Nitrogen Physisorption and DRIFTS

The liquid nitrogen physisorption revealed that synthesized NA-based polymers are mainly micro-mesoporous with predominant microporosity; SSA_BET_ varied from 757 m^2^/g up to 1065 m^2^/g (see Table 1 and Figure 2). It is noteworthy that with the increase in methylal amount from 15 mmol up to 120 mmol, the SSA_BET_ went through a maximum at 30 mmol (Figure 2d), while the relative micropore volume in the case of NA30 had the lowest value.

For this series of polymers (NA15–NA120), analysis by DRIFTS was carried out. Normalized IR-spectra are shown in Figure 3. It can be seen that for all the samples, the spectra are characterized by the absorption bands at 3062, 3042, and 3008 cm^−1^ belonging to stretching vibrations of C–H bonds in aromatic rings of NA. The absorption band at about 3650 cm^−1^ refers to vibrations of the OH–groups of the polymer and water, which did not form hydrogen bonds. A broad band with the maximum at about 3530 cm^−1^ corresponds to stretching vibrations of OH–groups belonging to carboxyl groups that formed hydrogen bonds. Bands at 1607 and 1500 cm^−1^ refer to stretching vibrations of C–C bonds in aromatic rings. The band at 1370 cm^−1^ likely refers to deformation vibrations of CH_3_–groups.

There are numerous absorption bands, which correspond to oxygen-containing species: bands at 1770, 1725, and 1700 cm^−1^ refer to stretching vibrations of >C=O belonging to keto- and carboxyl groups, namely Ar–O–C(O)–R, Ar–C(O)–O–R, and Ar–COOH [29,30]. For hyper-crosslinked polystyrene, Davankov et al. [31] suggested that the absorption bands in the region of 1740–1600 cm^−1^ can be attributed to hindered vibrations of C–C bonds and bond angles in benzene rings. However, in the case of NA-based polymers synthesized in this work, we can also observe several strong bands of stretching vibrations of C–O bonds belonging to ester and also ether groups, i.e., 1267 cm^−1^ (Ar–C(O)–O–R), 1190 cm^−1^ (Ar–O–C(O)–R), and 1105 cm^−1^ (Ar–O–CH_2_–R).

In the range of 1460–1420 cm^−1^, in-plane bending vibrations of CH_2_–groups belonging to Ar–CH_2_–Ar and Ar–CH_2_–OH can be found. If we take a closer look at the region of 2960–2800 cm^−1^ (Figure 3b), we will find several absorption bands of stretching vibrations of C–H bonds in CH_2_–groups belonging to Ar–CH_2_–Ar and Ar–CH_2_–OH, all of which are the result of methylal binding to the rings of NA.

The intense absorption band at 2924 cm^−1^ was chosen to determine if there is any dependence between its signal (in K-M units) and the amount of methylal used for the polymers’ synthesis (Figure 4).

As can be seen, the increase in methylal amount from 15 mmol up to 60 mmol results in a corresponding linear increase in its binding to NA monomers. It is noteworthy that the amount of bonded methylal does not reflect the degree of the polymer crosslinking (groups Ar–CH_2_–Ar between neighboring chains), since methylal also participates in the formation of oxygen-containing moieties. Additionally, SSA_BET_ decreased with the increase of methylal amounts higher than 30 mmol (see Figure 2d) in accordance with the data of DRIFTS.

When the amount of methylal is higher than 60 mmol, a relative decrease in the signal of the absorption band at 2924 cm^−1^ is seen; this can be due to an excessive amount of unreacted methylal. While adding the methylal to the NA-containing solution, fast gelation occurred, after which the mass transfer through the forming dense polymer layer was hindered, in spite of mixing, and the excessive amount of methylal could not penetrate into the reaction volume.

The functionalized polymers (NL60, NNA120, and SNA120) were also studied by liquid nitrogen physisorption and DRIFTS.

The polymer NL60 synthesized using NL as a monomer and 60 mmol of methylal revealed a nearly double decrease in the SSA_BET_ as compared with NA60—from 943 m^2^/L to 519 m^2^/L (see Table 1 and Figure 5a). Moreover, a noticeable shift in porosity was found (see Figure 5b, comparison of NA60 and NL60 polymers). The IR-spectra of NA60 and NL60 were rather similar (Figure 6a), with the only difference being that in the case of NL60, the signal of OH-groups was more intense.

After nitration and sulfonation of NA120 (samples NNA120 and SNA120), SSA_BET_ was also decreased, and the porosity changed in favor of meso/macro-pores (Figure 5). Nitration resulted in a nearly double decrease in the SSA_BET_ (Table 1), while after the sulfonation, only a 1.2-fold decrease in the SSA_BET_ was found.

As can be seen from the data of DRIFTS (Figure 6), both nitration and sulfonation resulted in the cleavage of methylene bridges (absorption band at 2924 cm^−1^) with simultaneous oxidation, especially in the case of nitration (note the increase in the number of oxygen-containing moieties).

#### 2.1.2. TGA

For the polymers NA120, NNA120, and SNA120, TGA was carried out. The results are presented in Figure 7.

The sample NA120 lost about 5% of its weight in the temperature range of 30–197 °C, which likely corresponded to the evaporation of the solvents from micropores. The intensive destruction of NA120 in argon medium was observed over 365 °C, and the residual mass was ca. 67.7% while reaching temperatures close to 600 °C.

In the case of SNA120, two small regions of weight loss were found: (i) in the range of 30–128 °C (likely due to the solvents’ evaporation); and (ii) in the range of 128–230 °C (corresponding to the elimination of sulfonate groups). The intensive destruction of SNA120 in argon medium was observed at temperatures above 366 °C, and the residual mass was ca. 75.6% at 600 °C.

NNA120 constantly lost weight without any obvious peaks in DTG curves. The intensive destruction of NNA120 started after 249 °C; the residual mass was ca. 74.3%.

#### 2.1.3. XPS

XPS revealed that all the synthesized polymers contain carbon, oxygen, and chlorine on the surface. Independently of the amount of methylal used for polymer synthesis, the content of oxygen and chlorine on the surface was in the range of 4–6 at.% and 2–4 at.%, respectively. The binding energy (BE) of Cl 2p_3/2_ was equal to 200.4 eV, which corresponded to chlorine bonded to aromatic rings of NA [32].

In the case of NNA120, the existence of nitro-groups was confirmed by the XPS data; the BE of N 1s was equal to 405.6 eV [32] (Figure 8a). The nitrogen content on the surface of the NNA120 was 2.0 at.%. On the surface of SNA120, sulfur (1.7 at.%) was found in the form of sulfonate groups; the BE of S2p_3/2_ was equal to 168.7 eV [32] (Figure 8b).

In spite of the fact that in some publications the formation of SO_2_-bridges between aromatic rings during the polymer sulfonation was stated [33,34,35], in the SNA120 sample only one state of sulfur corresponding to –SO_3_–R was found, where R is either H or organic C. It was reported elsewhere that the BEs of –SO_3_–H (168.3 eV) and –SO_3_–C (168.6 eV) are very close, while for C-SO_2_-C, the BE of S 2p_3/2_ is 167.6 eV [36].

### 2.2. Characterization of the Catalysts

Synthesized and reduced Pd-containing catalysts (1%-Pd/NA60-R, 1%-Pd/NA120-R, 1%-Pd/NNA120-R, 1%-Pd/SNA120-R, 1%-Pd/NL60-R) were analyzed by DRIFTS in order to find out if any changes occurred during the treatment with NaBH_4_. Normalized IR-spectra of the catalysts 1%-Pd/NA120-R and 1%-Pd/NL60-R in comparison with the IR-spectra of corresponding polymers are shown in Figure 9.

It can be seen that the treatment with NaBH_4_ had no visual effect on the polymers’ structure. The same was observed for 1%-Pd/NA60-R. The increase in intensity of the absorption bands at 2924 cm^−1^ (which corresponds to CH_2_–groups), at about 3500 cm^−1^ (which corresponds to OH–groups), and at 1730 cm^−1^ (which likely corresponds to ester groups Ar–C(O)–O–R) can be due to the procedure of catalysts’ preparation and reduction, i.e., the THF and the product of its hydrolysis could remain in the polymer pores.

In the IR-spectrum of 1%-Pd/SNA120-R (Figure 10b), the absorption bands are the same as for SNA120. However, similarly to 1%-Pd/NA120-R and 1%-Pd/NL60-R, the intensity of the absorption bands corresponding to CH_2_–, OH–, and ester groups increased. This could be explained by the strong adsorption of solvents (THF) on the polymer.

In the case of 1%-Pd/NNA120-R (Figure 10a), there were some noticeable differences in the corresponding IR-spectra: a new band at ca. 3390 cm^−1^ occurred that can be attributed to the stretching vibrations of N–H forming hydrogen bonds. Additionally, the absorption band at 1619 cm^−1^ appeared, which can be referred to deformation vibrations of N–H of amino-groups. Thus, during the formation of Pd NPs, partial reduction of the nitro-groups of NNA120 occurred.

STEM was carried out for the reduced Pd-containing catalysts (1%-Pd/NA60-R, 1%-Pd/NA120-R, 1%-Pd/NNA120-R, 1%-Pd/SNA120-R, and 1%-Pd/NL60-R).

Figure 11 represents the HAADF STEM image of 1%-Pd/NA60-R and the corresponding EDX mapping. As can be seen, a non-uniform distribution of palladium occurred, which was likely due to microporosity along with the relatively high hydrophobicity of NA60. Huge aggregates of polydisperse Pd NPs with a mean diameter of 16.5 ± 7.3 nm were formed after the reduction.

1%-Pd/NA120-R (Figure 12a) as well as 1%-Pd/NL60-R (Figure 12b) revealed more uniform Pd distribution as compared with 1%-Pd/NA60-R and lower diameters of Pd NPs (12.1 ± 4.1 nm for 1%-Pd/NA120-R and 9.9 ± 2.6 nm for 1%-Pd/NL60-R). In the case of 1%-Pd/NL60-R, the smaller diameter and better distribution of Pd NPs can be due to the relatively higher oxygen content (6.9 at.%) as compared with 1%-Pd/NA120-R (4.1 at.%).

The smallest Pd NPs (3.0 ± 1.3 nm) with the narrowest NP size distribution were found in the case of 1%-Pd/NNA120-R (Figure 12c) in spite of the meso-macroporous structure of the polymer (NNA120). In the case of 1%-Pd/SNA120-R, Pd NPs with a mean diameter of 7.5 ± 3.3 nm and a broad size distribution were formed after the reduction of Pd acetate in meso-macroporous SNA120 (Figure 12d), which were still smaller as compared with NPs formed in NA120-R and NL60-R.

Some correlation can also be found between the oxygen content on the surface of initial polymers and the mean diameter of Pd NPs formed after the reduction with sodium borohydride (Figure 13).

Thus, it can be concluded that high microporosity along with hydrophobicity are unfavorable characteristics of polymeric support in terms of the morphology of the resulting catalytic material.

The XPS study revealed that after the reduction treatment, palladium acetate was transformed to the metallic form. The BE of Pd 3d_5/2_ on the surface of the samples 1%-Pd/SNA120-R and 1%-Pd/NNA120-R was 335.8 eV, which could be attributed to Pd^0^ (67% and 32%, respectively, of all the Pd on the surface), and also 337.0–337.4 eV, which corresponded to PdO [32].

It is noteworthy that after the reduction of sulfonated polymer (sample 1%-Pd/SNA120-R), the state of sulfur remained the same as on the surface of the initial SNA120 (BE of S 2p_3/2_ was equal to 168.5 eV). In the case of 1%-Pd/NNA120-R, two states of nitrogen were found with BEs of N 1s equal to 405.9 eV (corresponding to the nitro-groups, 69% of all the nitrogen on the surface) and 399.7 eV (corresponding to the amino-groups, 31% of all the nitrogen on the surface), which indicated partial reduction of NO_2_-groups with NaBH_4_ in the presence of Pd acetate.

On the surface of 1%-Pd/NL60-R, palladium was found in the partially oxidized form PdO/Pd(PdO_x_, x ≥ 1), with BE of Pd 3d_5/2_ equal to 336.8 eV and PdO (BE is 337.4 eV). The share of PdO/Pd was about 64% of all the Pd on the surface; the share of PdO was 36%. In the case of 1%-Pd/NA120-R, palladium was found in the form of PdO/Pd(BE is 336.2 eV) and palladium acetate (BE is 338.2 eV) [32]. The share of PdO/Pd was about 72% of all the Pd on the surface, while Pd(CH_3_COO)_2_ was 28%. It is noteworthy that the BE value of 336.2 eV could also correspond to small Pd_n_ clusters [37]. However, since the data of STEM showed the formation of rather big Pd NPs (see Figure 12a), we believe that the surface of these particles was partially oxidized.

The BE of Pd 3d_5/2_ on the surface of the sample 1%-Pd/NA60-R was 336.8 eV, which could be attributed to partially oxidized Pd surface PdO/Pd (PdO_x_, x ≥ 1), and 339.2 eV, which likely corresponded to higher oxidation states of palladium (PdO_x_, x ≥ 2) [38].

### 2.3. Suzuki Cross-Coupling

All the synthesized catalysts were tested in a model reaction of Suzuki cross-coupling between BrAN and PBA under the following conditions: 1 mmol of BrAN, 1.5 mmol of PBA, 2.0 mmol of NaOH, 60 °C, 900 rpm, 0.2 mol.% of Pd. The results are presented in Table 2.

As can be seen from Table 2, all initial (unreduced) samples revealed higher activity as compared with the reduced ones. This can be due to the known fact that Suzuki cross-coupling proceeds via the formation of homogeneous Pd species that are present in unreduced samples, while in the case of preliminarily reduced catalysts, the dissolution of Pd NPs into the reaction media should take place first (see, for example, our recent study [39]). The selectivity with respect to the cross-coupling products (MBP) was nearly the same for all the catalysts about 97–98% (see Figure 14b and Figure 15b).

Among the unreduced catalysts, the highest activity was found for 1%-Pd/NA60, 1%-Pd/NNA120, and 1%-Pd/SNA120 (Table 2, Figure 14a). In the case of reduced samples, the highest activity was found for 1%-Pd/NL60-R (Table 2, Figure 15a).

For the catalyst 1%-Pd/SNA120, the reaction scope was studied at variations of aryl halides: 4-bromonitrobenzene (BrNB), 4-bromoaminobenzene (BrAB), 4-bromobenzaldehyde (BrAL), and 4-bromotoluene (BrTL). The highest activity (209.2 mol_BrAN_/(mol_Pd_·min)) and selectivity (yield of cross-coupling product was >99%) were found in the case of BrNB (Appendix A).

Surprisingly, for the preliminarily reduced catalysts, there was no correlation between the sizes of Pd NPs, polymers’ porosity, and the observed catalytic behavior at the first use. To verify if the support hydrophilicity influences the catalyst stability, all catalysts were tested in the second reaction run (Figure 16).

As can be seen from Figure 16, the catalysts based on the nitrogen- and sulfur-containing polymers revealed better stability as compared with the pure NA-based and NL-based samples. Earlier [40], we proposed that the higher hydrophilicity of the polymer can be responsible for better catalyst stability. Thus, for the NA-based polymers, the following trend can be found: the higher the oxygen content, the higher the remaining activity at the second reaction run (see Figure 17). Note that the remaining activity was estimated as a ratio (in %) of BrAN conversion achieved at the second run by 60 min to the initial conversion presented in Table 2.

In spite of the relatively higher stability of the nitrogen- and sulfur-containing polymers as compared with the other samples, one can see that the loss of activity is noticeable. The samples 1%-Pd/NNA120-R and 1%-Pd/SNA120-R were also tested in the third run and the conversion of BrAN by the end of the third run was 50% and 32%, respectively.

The catalyst behavior at multiple reuses in the case of cross-coupling reactions has a complex nature [40]—not only does the adsorption of reaction mixture components take place, but also the leaching of Pd and the formation of less active Pd species and NPs. Thus, for the samples based on the functionalized polymers (NL60, NNA120, and SNA120), additional STEM studies were carried out for both the initial (unreduced) (see Appendix A) and reduced catalysts (Appendix A) taken either after the second run or the third run. For unreduced samples, it was found that numerous Pd NPs were formed after the Suzuki reaction (Appendix A). Interestingly, the activity at the second run of the Suzuki reaction correlated with the sizes of the Pd NPs (Appendix A). In the case of preliminary reduced samples, the sizes of the Pd NPs were also estimated after the repeated runs (Appendix A). For example, for the 1%-Pd/NL60-R, the mean diameter of the Pd NPs changed slightly from 9.9 ± 2.6 nm to 9.4 ± 2.4 nm after the second use (see Appendix A). In the case of 1%-Pd/SNA120-R taken after the third run, the mean diameter of the Pd NPs decreased from 7.5 ± 3.3 nm to 5.6 ± 2.8 nm (Appendix A); while for 1%-Pd/NNA120-R, the size distribution of the Pd NPs was ruined and, along with the particles at the detection limit of STEM, Pd aggregates were found (Appendix A and Figure 7). Note that partial dissolution and reprecipitation of Pd NPs are also typical for cross-coupling processes [40]. Thus, the rate of dissolution of Pd NPs may also be responsible for the observed activity.

Pd leaching can be expected, considering the heterogenous/homogeneous mechanism of the Suzuki reaction. For the catalyst 1%-Pd/NNA120-R, the hot-filtration test was carried out (Appendix A). It was found that the conversion of BrAN using the filtrate increased from 7% to 14.4%, confirming the leaching process.

## 3. Discussion

In this work, for the first time, a series of NA-based polymers was synthesized at variations of the amount of crosslinking agent (methylal). It was shown that methylal not only participates in the formation of Ar–CH_2_–Ar bridges between neighboring NA molecules but also contributes in the formation of oxygen-containing moieties. Moreover, functionalization of the NA-based polymers with -OH, -SO_3_R, or –NO_2_ groups was carried out in order to alter their porosity and relative hydrophilicity.

Using different types of polymers (fully aromatic: NA60 and NA120), containing hydroxyl groups (NL60), sulfonate groups (SNA120), and nitro-groups (NNA120), palladium-containing catalysts were synthesized (1 wt.% of Pd) with Pd acetate as a precursor. It was found that NA-based HAPs can serve as a support for Pd NPs. In our previous work [21,39], we showed that small Pd NPs at 2–4 nm in diameter can be formed after reduction in a hydrogen flow at 300 °C of Pd-precursor on commercial hyper-crosslinked polystyrene. In this study, to demonstrate the effect of functional groups of the synthesized novel NA-based polymers on the formation of the metal NPs, we chose liquid-phase reduction under mild conditions (0–5 °C, NaBH_4_) to protect polymers’ functionalities (-OH, -SO_3_R, or -NO_2_ groups) from destruction. As a result, it was shown that the higher hydrophilicity of the polymeric support allows the formation of smaller Pd NPs with a narrower particle size distribution. The best results were obtained for the nitrated polymer (NNA120, where 120 refers to methylal content in mmoles): the diameter of Pd NPs was about 3 nm, and the initial micro-mesoporous polymer became the meso-macroporous one. Thus, it was concluded that in a liquid phase where Pd salt precursor has a tendency to migrate, the porosity is a minor factor influencing the NPs’ sizes as compared to the presence of surface functional groups.

Both the initial (unreduced) catalysts, containing Pd(II) species, and the nanoparticulate ones were tested in a model reaction of the Suzuki cross-coupling between BrAN and PBA under mild conditions (EtOH-H_2_O mixture (5:1) as a solvent, 60 °C, 0.2 mol.% of Pd). For the preliminarily reduced samples, no correlation between the sizes of Pd NPs and catalytic efficiency was found. This may be partially due to the known homogeneous/heterogeneous mechanism of the Suzuki reaction, but it may also indicate the influence of the support nature. Afterwards, each initial (unreduced) and reduced sample was tested in a second run of the Suzuki reaction. It was found that Pd catalysts based on functionalized (hydrophilic) polymers are more stable as compared with the relatively hydrophobic polymer (NA120) used as a support.

Earlier [40], we showed that the loss of catalytic activity can be due to numerous factors: leaching of Pd followed by its precipitation leading to the growth of Pd NPs, adsorption of the reaction products on Pd surface and on the support, etc. Therefore, the hydrophilicity of the polymeric support is also very important for the catalyst stability. The results presented in this work demonstrated that the introduction of hydrophilic substituents in HAPs may improve catalyst stability during cross-coupling reactions.

## 4. Materials and Methods

### 4.1. Materials

4-Bromoanisole phenylboronic acid (PBA, 95%), biphenyl (BP, 99.5%), *N*,*N*-dimethylformamide (DMF, 99.8%), diphenylamine (DPA, 99%), formaldehyde dimethylacetal (dimethoxymethane, methylal, 99%), methanol (MeOH, 99.8%), naphthalene (NA, 99%), 1-naphthol (NL, ≥99%), nitric acid (HNO_3_, 70%), sulfuric acid (H_2_SO_4_, 96%), tetrahydrofuran (THF, ≥99.9%), ethanol (EtOH, ≥99.8%), isopropanol (*i*-PrOH, ≥99.5%), FeCl_3_ (anhydrous, ≥99.99%), and sodium hydroxide (NaOH, ≥98%) were purchased from Sigma-Aldrich, St. Louis, MO, USA. 1,2-Dichloroethan (1,2-DCE, 99.9%) was purchased from JSC “1 Base Chemical Reagents” (Staraya Kupavna, Russia). Palladium acetate (Pd(CH_3_COO)_2_, 47.68%ofPd) were obtained from JSC “Aurat”, Moscow, Russia. All chemicals were used as received. Distilled water was purified with an Elsi-Aqua water purification system.

### 4.2. Synthesis and Characterization of Polymers

Synthesis of the NA-based polymers was carried out by one-step crosslinking of NA with formaldehyde dimethyl acetal (methylal) according to the procedure described elsewhere [41,42]. 1,2-DCE was used as a solvent and anhydrous FeCl3 as a polymerization catalyst. Briefly, 60 mmol of FeCl_3_ were placed in 20 mL of 1,2-DCE and mixed (300 rpm) till the dissolution. Then, NA (20 mmol) was added. After that, a certain amount of methylal (15–120 mmol) was slowly introduced into the reactor at constant stirring (1000 rpm), while the temperature did not exceed 42 °C. Afterwards, the temperature was raised up to 45 °C (maintained for 5 h), and then up to 80 °C (maintained for 19 h). The resulting polymer was washed with MeOH for 24 h and dried at 40 °C under vacuum for 24 h.

The obtained polymers were characterized by liquid nitrogen physisorption, diffuse reflectance infrared Fourier transform spectroscopy (DRIFTS), thermogravimetric analysis (TGA), and X-ray photoelectron spectroscopy (XPS).

Liquid nitrogen physisorption was carried out using a Beckman Coulter SA 3100 (Coulter Corporation, Miami, FL, USA). Prior to the analysis, each sample was placed in a quartz cell installed in the Becman Coulter SA-PREP. The samples were pretreated over 60 min under nitrogen at 120 °C. Once the pretreatment was completed, the cell was cooled and weighed, and then transferred to the analytical port. Analysis was performed at −196 °C and at relative pressure of 0.9814 (for pores less than 100 nm in diameter) to obtain a PSD (ADS) profile.

DRIFTS was carried out using an IRPrestige-21 FTIR spectrometer (Shimadzu, Kyoto, Japan) equipped with a DRS-8000 diffuse reflectance accessory (Shimadzu, Kyoto, Japan). The background sample was a mirror of the material of the optical system of the DRS-8000 accessory. All spectra were recorded in the 4000–500 cm^−1^ range of wavenumbers at a resolution of 4 cm^−1^.

TGA was carried out using scales TG 209 F1 Iris (Netzsch, Selb, Germany). Argon was used as a medium and as a protective gas. The following temperature program was used: 40 °C (5 min) then heating up to 600 °C (10 °C/min) and then 600 °C for 10 min. Flow rates of argon through the furnace of TG scales and also the protective flow rate was equal to 20 mL/min.

XPS data were obtained using Mg Kα (hν = 1253.6 eV) radiation with an ES-2403 spectrometer (Institute for Analytic Instrumentation of RAS, Saint Petersburg, Russia) equipped with an energy analyzer PHOIBOS 100-MCD5 (SPECS, Berlin, Germany) and X-ray source XR-50 (SPECS, Berlin, Germany). All the data were acquired at an X-ray power of 250 W. Survey spectra were recorded at an energy step of 0.5 eV with an analyzer pass energy of 40 eV. High-resolution spectra were recorded at an energy step of 0.05 eV with an analyzer pass energy of 7 eV. Samples were outgassed for 180 min before analysis and were stable during the examination. The data analysis was performed via Casa XPS. Binding energies (BEs) were determined with the error ±0.1 eV.

### 4.3. Modification of Polymers with Niro- and Sulfo-Groups

Since the 1-nitronaphthalene did not undergo crosslinking, the nitro-groups were introduced by the nitration. Nitration of NA120 (Figure 18a) was carried out according to the procedure described elsewhere [43]. 

In a typical experiment, 0.5 g of the polymer was mixed with 5 mL of DMF. The suspension was leaved for 1 h at 25 °C and constant stirring (200 rpm). Then, the suspension was placed in the ice bath (0–5 °C) and purged with argon. After that, 20 mL of preliminarily cooled concentrated HNO_3_ was slowly added and then 5 mL of concentrated H_2_SO_4_ was added dropwise. The mixture was then placed in an oil bath at 50 °C and 1000 rpm for 4 h. Finally, the suspension of the nitrated polymer (NNA120) was carefully poured out in the pure ice. After the melting of ice, the aqueous suspension of NNA120 was filtered under vacuum through the paper filter, and the precipitate was thoroughly washed with H_2_O (ca. 1 L), and finally with EtOH. It is noteworthy that the first portions of EtOH extracts contained small amounts of yellowish, low-molecular weight nitration products. The resulting polymer, NNA120, was dried for 24 h at 65 °C (Figure 18b).

Sulfonation of NA120 was carried out according to the procedure described elsewhere [33,44,45]. In a typical experiment, 0.5 g of the polymer was mixed with 5 mL of 1,2-DCE. The suspension was leaved for 1 h at 25 °C and constant stirring (200 rpm). Then, 25 mL of concentrated H_2_SO_4_ was added, and the mixture was heated up to 80 °C and leaved for 4 h at 1000 rpm. Then, the suspension was placed in the ice bath, and the small portions of diluted H_2_SO_4_ (1:10; 1:20; 1:40; and 1:80) were slowly added. Finally, pure water was added. The suspension of SNA120 was filtered under vacuum through the paper filter; the precipitate was thoroughly washed with H_2_O (ca. 1 L) and finally with EtOH. The resulting polymer, SNA120, was dried for 24 h at 65 °C (Figure 18c).

### 4.4. Catalyst Synthesis and Characterization

Catalysts were synthesized via the wet-impregnation method. In a typical experiment, 0.5 g of the polymer was impregnated with 3 mL of the THF solution of precursor (Pd(CH_3_COO)_2_) of a chosen concentration. Then, the sample was air-dried at 65 °C until a constant weight was achieved. Thus, the following catalysts were synthesized and designated: 1%-Pd/NA60 (1.4 wt.% of Pd confirmed by the XFA), 1%-Pd/NA120 (1.1 wt.% of Pd), 1%-Pd/NNA120 (1.0 wt.% of Pd), 1%-Pd/SNA120 (1.0 wt.% of Pd), and 1%-Pd/NL60 (1.0 wt.% of Pd). These catalysts were reduced with NaBH_4_ and were designated as 1%-Pd/NA60-R, 1%-Pd/NA120-R, 1%-Pd/NNA120-R, 1%-Pd/SNA120-R, and 1%-Pd/NL60-R. The reduction was carried out by the slow dripping of NaBH_4_ solution (0.1 mol/L, total volume 50 mL) to the suspension of a catalyst (0.06 g in 20 mL of EtOH) at 0–5 °C. After the hydrogen release was finished, the suspension was filtered under vacuum through the membrane filter (Nylon, 0.22 μm pore size, 50 mm diameter), thoroughly washed with H_2_O and EtOH, and dried for 24 h at 65 °C.

Note that all the synthesized catalyst samples (both the initial and reduced ones) were stored in air.

The catalysts were characterized by DRIFTS, XPS, and scanning transmission electron microscopy (STEM).

STEM characterization was carried out using a FEI Tecnai Osiris instrument (Thermo Fisher Scientific, Waltham, MA, USA) operating at an accelerating voltage of 200 kV, equipped with a high-angle annular dark field (HAADF) detector (Fischione, Export, PA, USA) and an energy-dispersive X-ray (EDX) microanalysis spectrometer (EDAX, Mahwah, NJ, USA). Samples were prepared by embedding them in epoxy resin followed by microtoming (ca. 50 nm thick) at ambient conditions. For the image processing, Digital Micrograph (Gatan, Pleasanton, CA, USA) software and TIA (Thermo Fisher Scientific, Waltham, MA, USA) were used. A Holey carbon/Cu grid was used as a sample support.

### 4.5. Reaction Procedure and Analysis of the Reaction Mixture

The Suzuki cross-coupling was carried in a temperature-controlled glass batch reactor with a magnetic stirrer at the following conditions: 900 rpm, 1 mmol of BrAN, 1.5 mmol of PBA, 2.0 mmol of NaOH, 0.2 mol.% of Pd (with respect to BrA) at 60 °C. All experiments were carried out in air using an EtOH-H_2_O mixture (volumetric ratio 5:1) as a solvent. The total volume of the liquid phase was 30 mL. The choice of the EtOH-H_2_O mixture and NaOH was based on our preliminarily studies [46]. 

Before the catalyst addition into the reactor, in each experiment, a blank test (duration of 60 min) was carried out in order to ensure that there is no reaction without the catalyst since even small amounts of leached palladium, if remaining on the stir bar or on the reactor walls, can catalyze the cross-coupling [47]. In all experiments, PBA was used in excess with respect to BrAN due to the possible non-selective PBA homo-coupling with the formation of BP.

In each catalytic experiment, samples of the reaction mixture were periodically taken and analyzed via GC-MS (Shimadzu GCMS-QP2010S) equipped with a capillary column HP-1MS (100 m × 0.25 mm i.d., 0.25 μm film thickness). Helium was used as a carrier gas at a pressure of 74.8 kPa and linear velocity of 36.3 cm/s. The oven temperature was programmed: 120 °C (0 min) → 10 °C/min (160 °C) → 25 °C/min (300 °C) → 300 °C (2.4 min). The temperature of the injector, interface, and ion source was at 260 °C, ranging from 10 up to 500 *m*/*z*. The concentrations of the reaction mixture components were calculated using the internal standard calibration method (DPA was used as an internal standard). It is noteworthy that during the GS-MS analysis, PBA underwent dehydration and trimerization, which affected the signal intensity. Due to the low level of confidence in the quantitative analysis of PBA, its consumption was not monitored.

Catalytic activity was characterized by the initial transformation rate, *R_0_*, defined as the tangent of the slope of the initial linear part on the kinetic curves of BrAN consumption, and related to the amount of palladium in the system: *R*_0_ = (*N_BrAN_*_,0_ *– N_BrAN_*_,*i*_) × τ*_i_*^−1^ × *N_Pd_*^−1^, where *N_BrAN_* is the number of moles of BrAN transformed by the reaction time *τ*; *N_Pd_* is the overall number of moles of Pd; and *τ* is the reaction time, in minutes.

The conversion (*X*,%) of BrAN was defined as *X*,% = (*N_BrAN_*_,0_ − *N_BrAN_*_,*i*_) × *N_BrAN_*_,0_^−1^ × 100.

The selectivity(*S*,%) with respect to MBP was defined as a share of MBP among the reaction products (BP and MBP): *S*,% = *N_MBP,i_* × (*N_MBP,i_*+ *N_BP,i_*)^−1^× 100.

The yield (Y,%) of MBP was defined as *Y*,% = *S,*% *× X,*% × 0.01.

### 4.6. Catalysts Separation from the Reaction Mixture for Reuse

After the reaction completion, the catalysts were filtered under vacuum using a membrane filter (Nylon, 0.22 μm pore size, 50 mm diameter) and sequentially washed with EtOH (30 mL), *i*-PrOH (30 mL), water (500 mL), and EtOH (30 mL). Then, they were dried until a constant weight at 65 °C. It is noteworthy that for the repeated use, several (at least 2) catalyst samples were collected from the previous reaction runs and the averaged catalyst sample was taken for the next run. In this way, all the reaction conditions remained unchanged, including the catalyst weight.

### 4.7. Procedure of Hot-Filtration of the Reaction Mixture

The reaction was started as usual according to the procedure described in Section 4.5. After 5 min, 5 mL of the reaction mixture was separated (start of hot-filtration) using a syringe equipped with membrane (PTFE, 0.22 μm pore size) and promptly transferred to the second glass batch reactor, which was preliminarily thermostated and sealed. After that, the filtrate was kept under the reaction conditions for 60 min with following sampling and analysis.

## 5. Conclusions

In this study, we showed that NA-based HAPs can be easily synthesized by one-stage Friedel–Crafts crosslinking. NA units can be functionalized with -OH, -SO_3_H, and -NO_2_ groups, and the synthesized HAPs can be used as supports for Pd-containing catalysts active in Suzuki cross-coupling. All the NA-based polymers contain about 4–6 at.% of the oxygen on the surface. Introduction of oxygen-containing functionalities improves the distribution and sizes of Pd NPs. Moreover, these functionalities can increase surface hydrophilicity, improving catalytic stability of Pd-HAPs in the Suzuki reaction.

Synthesized NA-based HAPs cannot fully prevent Pd leaching and the loss of catalytic activity at reuses due to the homogeneous/heterogeneous mechanism of the Suzuki reaction. However, we believe that the presented data can be useful for a better understanding of structure–properties relationships for polymer-based Pd catalysts and may find applications in other processes.

## Figures and Tables

**Figure 1 molecules-28-04938-f001:**
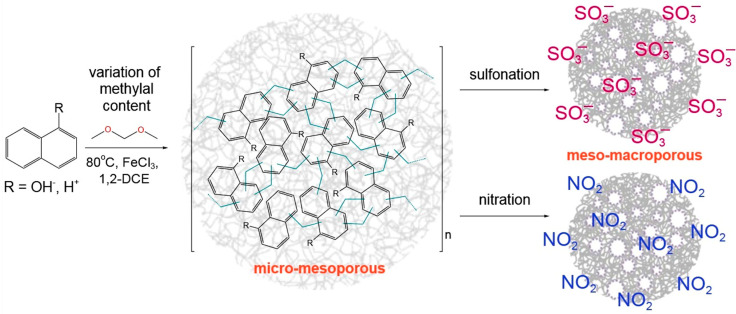
General scheme of the synthesis of NA-based polymers and their further modifications.

**Figure 2 molecules-28-04938-f002:**
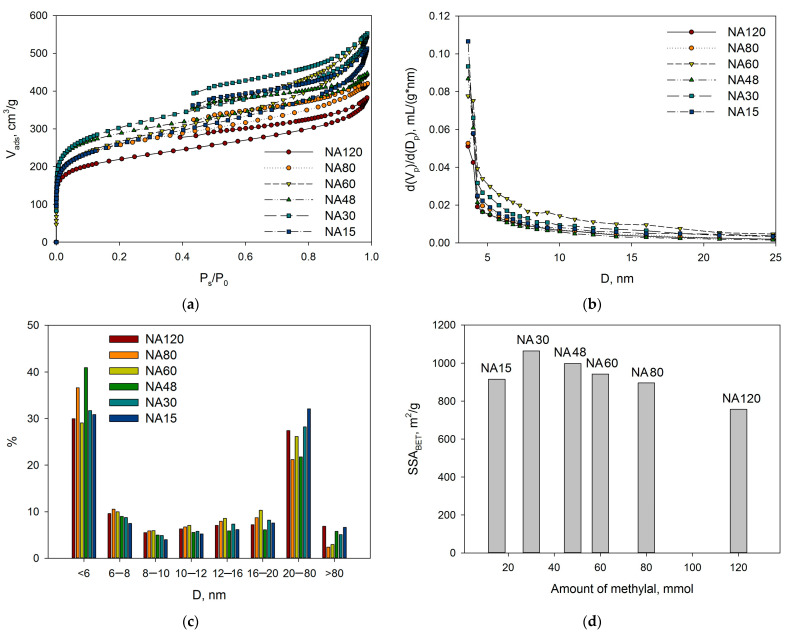
The results of low-temperature nitrogen physisorption for NA-based polymers obtained at variations of methylal amount: adsorption-desorption isotherms (**a**); pore volume distributions (**b**); relative pore size distributions (**c**); dependence of SSA_BET_ on the amount of methylal (**d**).

**Figure 3 molecules-28-04938-f003:**
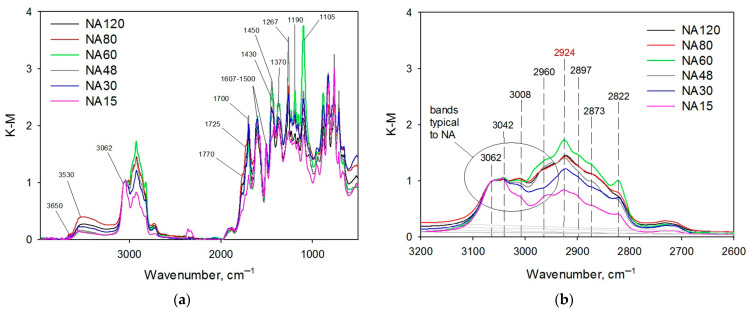
Normalized IR-spectra of NA-based polymers obtained at variations of methylal amount (**a**); scaled up region of the polymers’ spectra in the range of 3200–2600 cm^−1^ (**b**).

**Figure 4 molecules-28-04938-f004:**
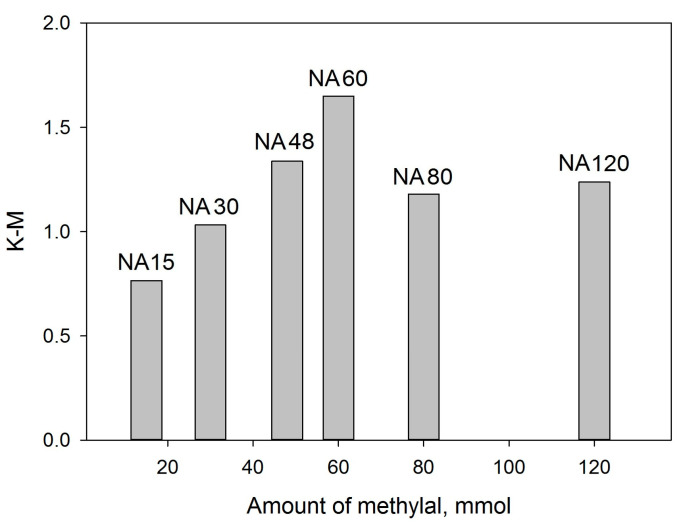
Dependence of the intensity of the band at 2924 cm^−1^ on the amount of methylal.

**Figure 5 molecules-28-04938-f005:**
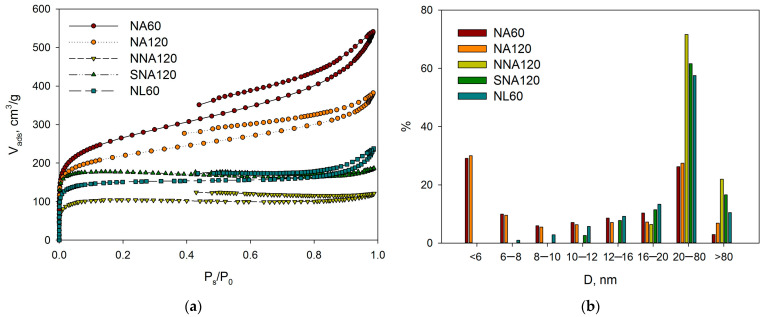
Influence of the polymer functionalization on the adsorption-desorption isotherms (**a**) and relative pore size distributions (**b**).

**Figure 6 molecules-28-04938-f006:**
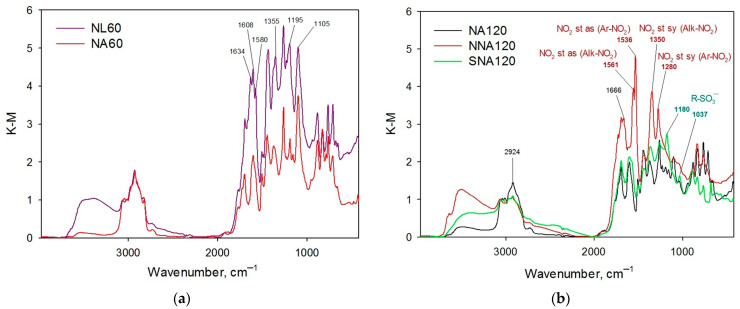
Comparison of the normalized IR-spectra:NL60 and NA60 (**a**); NNA120, SNA120, and NA120 (**b**).

**Figure 7 molecules-28-04938-f007:**
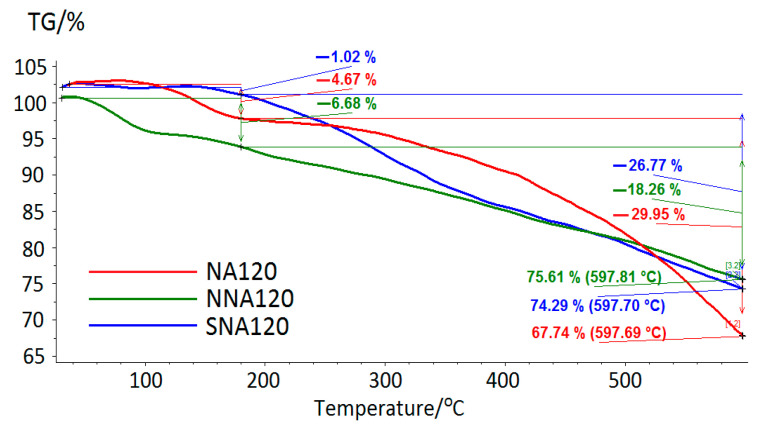
TG curves of NA120, NNA120, and SNA120.

**Figure 8 molecules-28-04938-f008:**
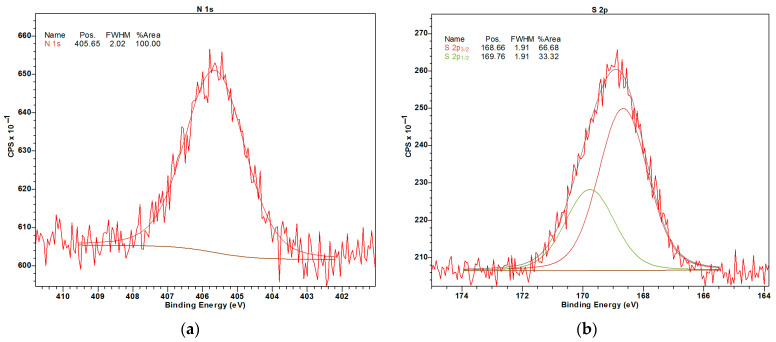
High-resolution XPS spectra of N 1s in NNA120 (**a**) and S 2p in SNA120 (**b**).

**Figure 9 molecules-28-04938-f009:**
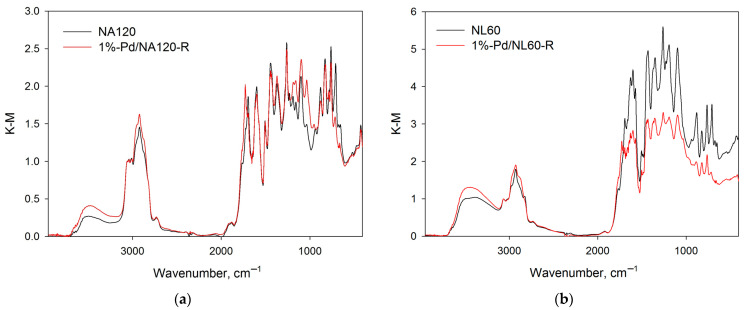
Comparison of the normalized IR-spectra: NA120 and 1%-Pd/NA120-R (**a**); NL60 and 1%-Pd/NL60-R (**b**).

**Figure 10 molecules-28-04938-f010:**
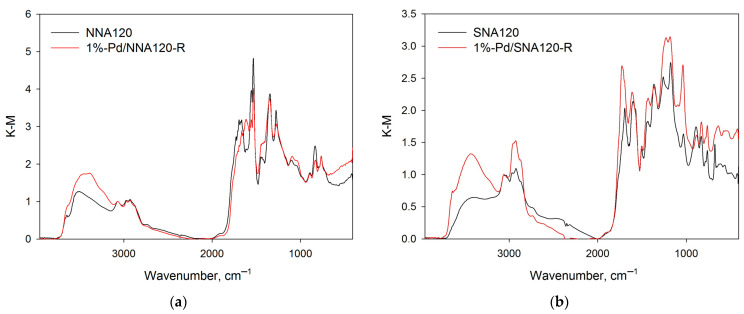
Comparison of the normalized IR-spectra: NNA120 and 1%-Pd/NNA120-R (**a**); SNA120 and 1%-Pd/SNA120-R (**b**).

**Figure 11 molecules-28-04938-f011:**
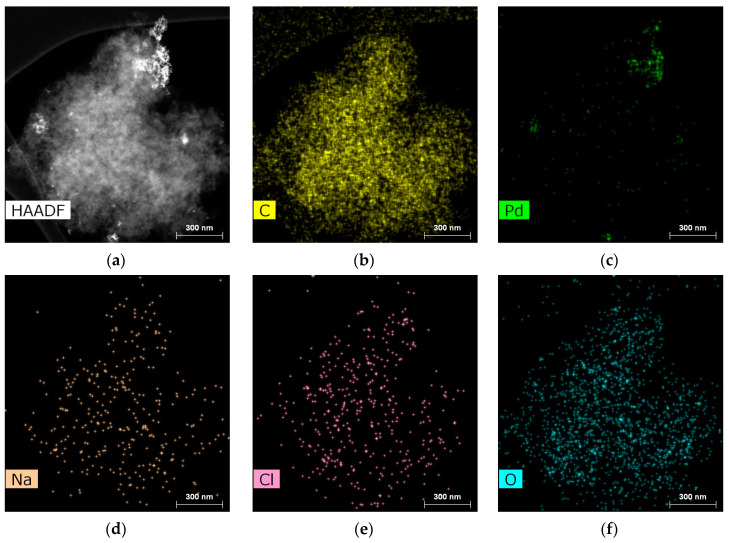
HAADF STEM image (scale 300 nm) of 1%-Pd/NA60-R(**a**) and EDX mapping of C (**b**), Pd (**c**), Na (**d**), Cl (**e**), and O (**f**).

**Figure 12 molecules-28-04938-f012:**
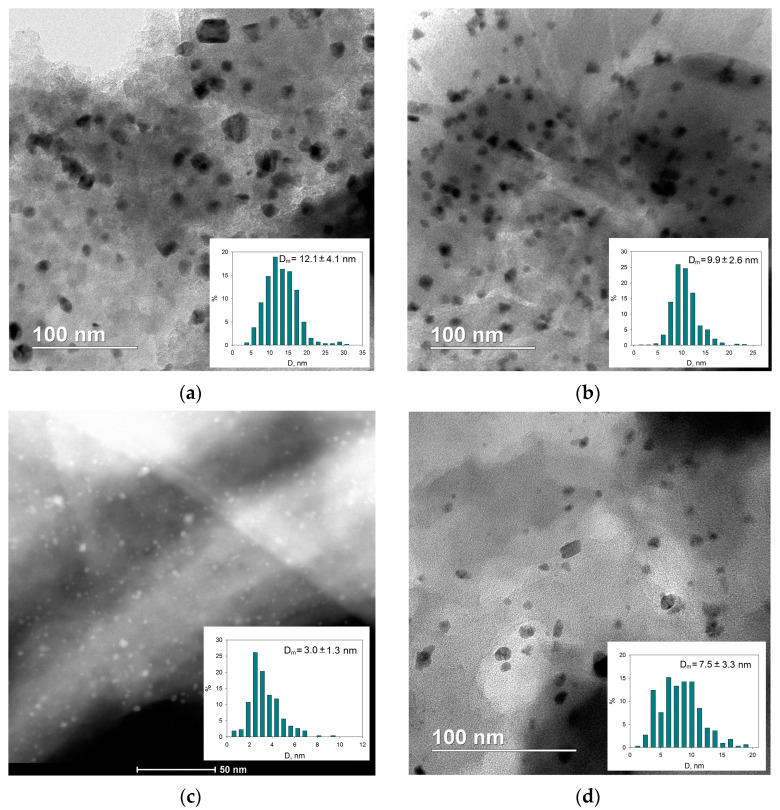
Bright-field STEM images of the reduced samples1%-Pd/NA120-R (**a**), 1%-Pd/NL60-R (**b**), 1%-Pd/SNA120-R (**d**) and HAADF STEM image of 1%-Pd/NNA120-R (**c**) (scale 50 nm).

**Figure 13 molecules-28-04938-f013:**
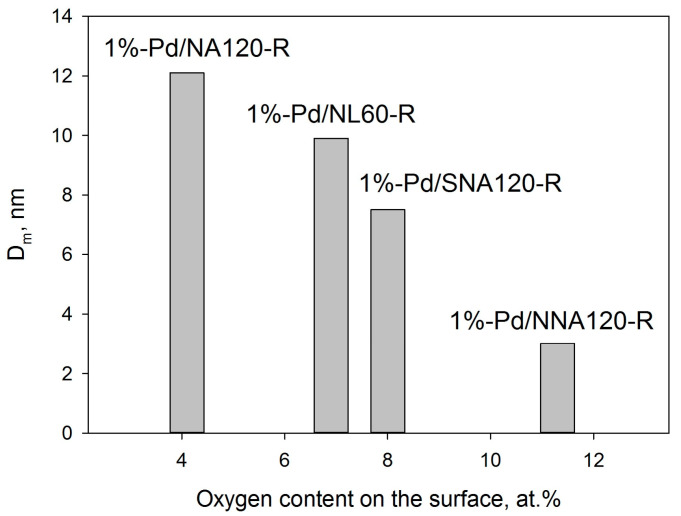
Dependence of the mean diameter of Pd NPs on the oxygen content (XPS data) on the surface of the initial polymers.

**Figure 14 molecules-28-04938-f014:**
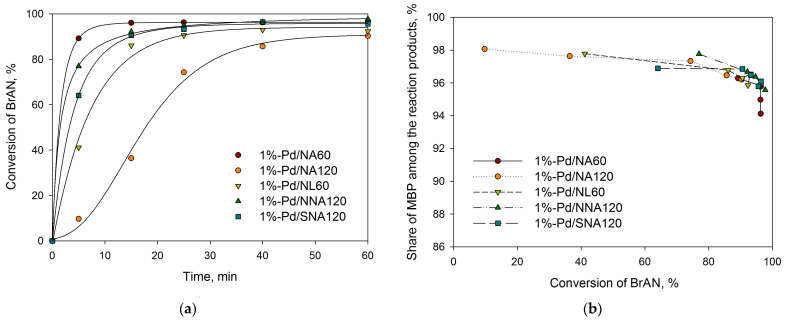
Dependence of BrAN conversion on time (**a**) and “selectivity” with respect to MBP vs. BrAN conversion (**b**) for the initial unreduced catalysts; reaction conditions: 1 mmol of BrAN, 1.5 mmol of PBA, 2.0 mmol of NaOH, 60 °C, 900 rpm, 0.2 mol.% of Pd.

**Figure 15 molecules-28-04938-f015:**
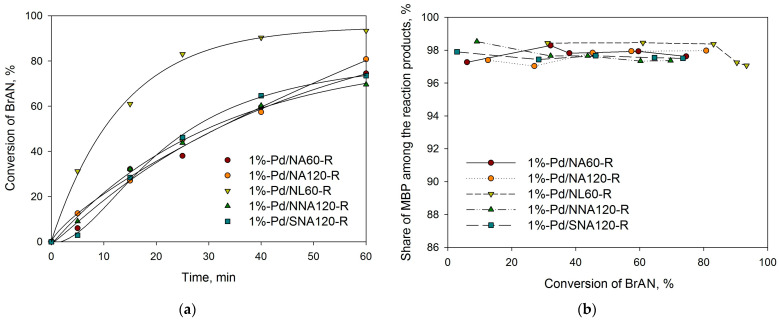
Dependence of BrAN conversion on time (**a**) and “selectivity” with respect to MBP vs. BrAN conversion (**b**) for the reduced catalysts; reaction conditions: 1 mmol of BrAN, 1.5 mmol of PBA, 2.0 mmol of NaOH, 60 °C, 900 rpm, 0.2 mol.% of Pd.

**Figure 16 molecules-28-04938-f016:**
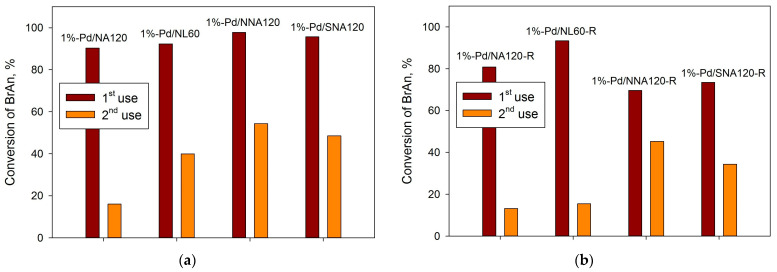
Study of catalysts’ stability in the second use for the initial catalysts (**a**) and reduced catalysts (**b**); reaction conditions: 1 mmol of BrAN, 1.5 mmol of PBA, 2.0 mmol of NaOH, 60 °C, 900 rpm, 0.02 g of catalyst.

**Figure 17 molecules-28-04938-f017:**
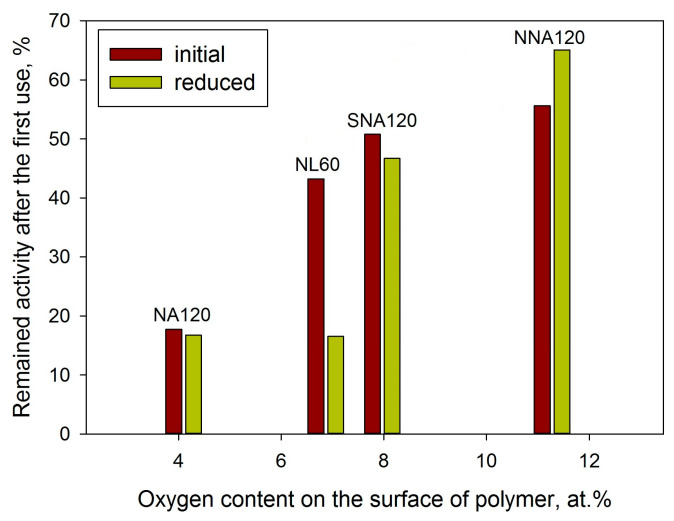
Dependence of the remained activity in the second reaction run on the oxygen content (XPS data) on the surface of the initial polymers.

**Figure 18 molecules-28-04938-f018:**
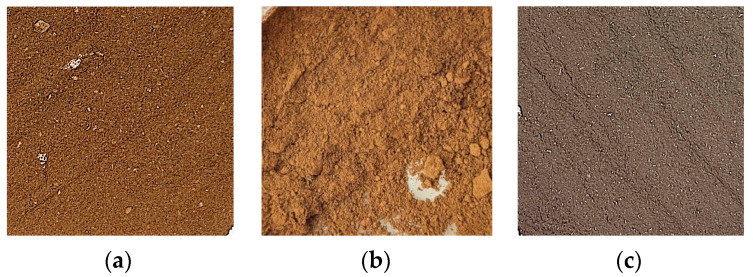
Appearance of the polymers: NA120 (**a**), NNA120 (**b**), and SNA120 (**c**).

**Table 1 molecules-28-04938-t001:** SSA and micropore volume of the NA-based polymers.

Sample	SSA_BET_, m^2^/g	SSA_t-plot_, m^2^/g	V_micropore_, mL/g
NA120	757	253 ^1^; 524 ^2^	0.234
NA80	896	365 ^1^; 553 ^2^	0.238
NA60	943	400 ^1^; 544 ^2^	0.246
NA48	998	281 ^1^; 743 ^2^	0.331
NA30	1065	603 ^1^; 462 ^2^	0.204
NA15	915	360 ^1^; 555 ^2^	0.254
NNA120	361	N/A	N/A
SNA120	608	N/A	N/A
NL60	519	94 ^1^; 425 ^2^	0.191

^1^ external SSA; ^2^ SSA of micropores.

**Table 2 molecules-28-04938-t002:** Results of cross-coupling for the catalysts containing Pd supported on the synthesized NA-based polymers.

Sample	Conversion of BrAN, %	Yield of MBP, %	*R*_0_, mol_BrAN_/(mol_Pd_·min)
1%-Pd/NA60	96.3	90.7	89.2
1%-Pd/NA120	90.3	86.9	14.9
1%-Pd/NL60	92.3	88.5	41.1
1%-Pd/NNA120	97.7	93.4	77.0
1%-Pd/SNA120	95.6	91.6	64.0
1%-Pd/NA60-R	74.5	72.7	9.3
1%-Pd/NA120-R	80.8	79.1	12.6
1%-Pd/NL60-R	93.4	90.6	31.2
1%-Pd/NNA120-R	69.6	67.9	10.7
1%-Pd/SNA120-R	73.5	71.7	9.5

## Data Availability

Data sharing is not applicable to this article.

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
