# Peer review of "Naphthalene-Based Polymers as Catalytic Supports for Suzuki Cross-Coupling"

_molecules, 2023, doi:10.3390/molecules28134938_

Round 1
Reviewer 1 Report
This manuscript describes the synthesis of naphthalene(NA)-based polymers via one-stage Friedel-Crafts crosslinking. I believe the manuscript is suitable for publication and some minor flaws must be fixed before consideration.
1. Author must provide a conclusion section with the main findings of the approach.
2. Some more examples of Suzuki coupling need to be included with selective catalyst.
3. Some images need to be improved and typos such as space issues and language must be polished again.
language need to be improved and typo errors should be minimized accordingly
Author Response
Response to Reviewer 1
Comments:
This manuscript describes the synthesis of naphthalene (NA)-based polymers via one-stage Friedel-Crafts crosslinking. I believe the manuscript is suitable for publication and some minor flaws must be fixed before consideration.
Point 1: Author must provide a conclusion section with the main findings of the approach.
Response 1: Agree.
Action taken: Conclusions were added.
Point 2: Some more examples of Suzuki coupling need to be included with selective catalyst.
Response 2: Agree.
Action taken: Additional experiments were carried out with different substrates over the sample 1%-Pd/SNA120. The highest activity (100% conversion within 5 min) at almost full yield (selectivity >99%) were found in the case of cross-coupling of 4-bromonitrobenzene. This information was added to the manuscript and included in Supplementary Materials.
Point 3: Some images need to be improved and typos such as space issues and language must be polished again.
Response 2: Agree.
Action taken: Language was corrected. Figures 1 and 7 were improved.
Reviewer 2 Report
The paper by Prof. Kiwi-Minsker and Prof. Nikoshvili reports the synthesis of naphthalene (NA)-based polymers by one-step Friedel-Crafts crosslinking used as a carrier for the formation of Pd NPs. The effect of different substituents on the morphology and performance of the newly obtained catalyst was studied. The catalysts were tested in a Suzuki reaction under mild reaction conditions. These new catalysts have demonstrated high efficiency with over 95% 4-bromoanisole conversion and high selectivity in a first cross coupling, but performance decreases as early as the second reuse. Therefore, the article deserves publication in Molecules, provided that the following points are clarified:
- The authors showed STEM images of the catalyst only before its use. The distribution of nanoparticles should also be monitored after their use in catalysis. This would help understand why the catalyst deactivates. Also, perhaps it is better to study metal nanoparticles directly with SEM.
- It would also be appropriate to add some tests on your organic solution after the synthesis. Is it possible that it penetrates into solution?
- By combining TGA, DTG and DTA some additional information can be obtained which can help to understand the behavior of the polymer during thermal heating.
- Has the molecular weight of the polymer been measured?
- Figure 1 should be clearer.
- A legend with the meaning of all acronyms and abbreviations used should be inserted at the end of the document.
- A recent application of polymer-supported organometallic catalysts in organic synthesis that could be mentioned in the introductory section is : A.M. Fiore, G. Varvaro, E. Agostinelli, A. Mangone, E. De Giglio, R. Terzano, I. Allegretta, M. M. Dell'Anna, S. Fiore, P. Mastrorilli, Eur. J.Inorg. chem. 2022, e202100943].
Author Response
Response to Reviewer 2
Comments
The paper by Prof. Kiwi-Minsker and Prof. Nikoshvili reports the synthesis of naphthalene (NA)-based polymers by one-step Friedel-Crafts crosslinking used as a carrier for the formation of Pd NPs. The effect of different substituents on the morphology and performance of the newly obtained catalyst was studied. The catalysts were tested in a Suzuki reaction under mild reaction conditions. These new catalysts have demonstrated high efficiency with over 95% 4-bromoanisole conversion and high selectivity in a first cross coupling, but performance decreases as early as the second reuse. Therefore, the article deserves publication in Molecules, provided that the following points are clarified.
Point 1: The authors showed STEM images of the catalyst only before its use. The distribution of nanoparticles should also be monitored after their use in catalysis. This would help understand why the catalyst deactivates. Also, perhaps it is better to study metal nanoparticles directly with SEM.
Response 1: agree.
Action taken: Since most of particles are buried into the polymers’ volume, it is better to provide microscopic study in transmission mode. For the samples based on the functionalized polymers (NNA120, SNA120, NL60) additional STEM studies were carried out for both the reduced and unreduced catalysts taken either after the second run or the third run. Corresponding information and discussion were added to the revised manuscript and included in Supplementary Materials.
Point 2: It would also be appropriate to add some tests on your organic solution after the synthesis. Is it possible that it penetrates into solution?
Response 2: Agree.
Action taken: We carried out hot-filtration test for the sample 1%-Pd/NNA120-R, which provided better stability among the synthesized samples. Corresponding information and discussion were added to the revised manuscript and included in Supplementary Materials.
Point 3: By combining TGA, DTG and DTA some additional information can be obtained, which can help to understand the behavior of the polymer during thermal heating
Response 3: There were almost no observed effects in DTG curves of the polymers. It should be noted that the temperatures of catalyst reduction and reaction are far from the region of the polymer thermal destruction.
Action taken: No
Point 4: Has the molecular weight of the polymer been measured?
Response 4: Unfortunately, we have no means to measure it.
Action taken: No
Point 5: Figure 1 should be clearer.
Response 5: Agree.
Action taken: Figure 1 was corrected.
Point 6: A legend with the meaning of all acronyms and abbreviations used should be inserted at the end of the document.
Response 6: Agree.
Action taken: List of abbreviations was added.
Point 7: A recent application of polymer-supported organometallic catalysts in organic synthesis that could be mentioned in the introductory section is: A.M. Fiore, G. Varvaro, E. Agostinelli, A. Mangone, E. De Giglio, R. Terzano, I. Allegretta, M. M. Dell'Anna, S. Fiore, P. Mastrorilli, Eur. J.Inorg. chem. 2022, e202100943].
Response 7: Agree.
Action taken: The reference was added.
Round 2
Reviewer 2 Report
The authors have integrated the paper which can be published.